# Estimating the effectiveness of national health insurance in covering catastrophic health expenditure: Evidence from South Korea

**Hyunwoo Jung**[1☯¤a], **Junhyup Lee**[2☯¤b]*

1 Department of Health Administration, Graduate School·BK21 Graduate Program of Developing global Experts in Health Policy and Management, Yonsei University, Wonju, South Korea, 2 Department of Public Health Science, Korea University, Seoul, South Korea

☯ These authors contributed equally to this work.
¤a Current address: Changjo Hall, Yonsei University Mirae Campus, Maeji-ri, Wonju-si, Gangwon-do, South Korea
¤b Current address: Hana Science Hall B-dong, Korea University, Anam-Dong, Sungbuk-Ku, Seoul, South Korea
* junhlee@korea.ac.kr

**Data Availability Statement:** The Korea Health Panel Study (KHPS) data we used in our study is de-identified data, and the information of all study subjects is anonymized. In addition, KHPS is reviewed annually by the institutional review board

## Abstract

The catastrophic health expenditure (CHE) indicator has been used to measure the medical cost burden of households. Many countries have institutionalized their health insurance systems to reduce out-of-pocket payments, the main contributor to the financial burden. However, there is no method to estimate how the insurance coverage reduces the CHE. This study proposes an approach to evaluate the effectiveness of insurance in reducing the CHE impacts in terms of incidence and gap, which are based on a modified calculation method of CHE. Additionally, we apply these methods to data from the Korea Health Panel Survey (2011–2016). The results are as follows. First, under the setting of a threshold of 10%, the CHE incidence rate was 19.26% when the Korean national health insurance benefits reduced the CHE's incidence for 15.17% of the population in 2017. Second, the results of the concentration index of CHE showed that the intensity approach of CHE is better than the incidence approach. Third, the new approach we applied revealed that health insurance reduces the burden of CHE to some degree, although it was not an efficient way to reduce CHE. In conclusion, this study provides new policy approaches to save the finances of national health insurance and reduce the intensity of CHE at the same time by raising the low-cost burden of medical services and lowering that of high cost. Moreover, we suggest that policymakers should focus on income level of the households rather than specific diseases.

## Introduction

In general, there are two approaches for estimating health insurance coverage: the first measures the share of the payment from public sources to the total health expenditure [1], while

(IRB) of the Korea Institute For Health And Social Affairs (KIHASA). Therefore, we don't think there will be a big ethical/legal issue in the data itself. However, KHPS is concerned about data leakage and sharing among unauthorized people. And KHPS requires to submit 'Consent for Data Utilization' when downloading data. This consent restricts the purpose of using the data for research purposes only and requires that individuals and institutions not use it for profit. It also contains provisions not to lend or transfer data to others or other entities. Therefore, we recommend you visit the homepage of KHPS, the institutional body (https://www.khp.re.kr), or contact the KHPS person in charge of data access. The mail address is khp@kihasa.re.kr.

**Funding:** The authors received no specific funding for this work.

the second uses the catastrophic health expenditure (CHE) indicator [2, 3]. The former calculates the ratio of public spending to the total health expenditure of the entire population, which has the advantage of providing a macroscopic view of how much of the total health care costs are covered by the government (including national health insurance). However, as this measurement only considers medical expenses and not individuals' financial burdens, it is difficult to estimate the medical costs' contribution to the economic burden on people at the individual or household level. To address this disadvantage, the CHE index is commonly used. The CHE indicator is computed as the proportion of out-of-pocket (OOP) payments to household income [3, 4].

Many relevant policy reports and articles about CHE are already published [2–8]. In particular, the World Health Organization (WHO) selected "fairness in financial contribution" as a goal to be achieved in health care systems and presented the CHE index as a method to estimate it [8, 9]. This indicator is not only used in macro-level studies on health coverage across many countries but also at the individual and household levels to analyze the causes of high medical expenditures and their impacts on household economies and poverty [10–14].

However, the CHE index has some limitations in estimating an insurance system's effectiveness in reducing people's economic burden because it uses only two variables (i.e., income and OOP). Therefore, this study proposes new methods to estimate health insurance systems' effectiveness in reducing the CHE's impacts on people by using and modifying the CHE calculation method. This new approach can provide insights into the medical cost burden on individuals from the consumer perspective and the level of health insurance coverage from the insurer and national perspectives. From these points of view, it is possible to reconsider how the national health insurance finance can "efficiently" lower the burden of medical expenses for the people. "Efficiency" is an important theme in the operation of national health insurance. The more benefits there are to health insurance, the more funding they require, which is composed of premiums [15, 16]. In other words, the irony arises because the tax burden of other general citizens must be raised to lower the burden of medical expenses on patients. Rather than expanding health insurance coverage by recklessly raising premiums and taxes, we believe that health insurance's policy direction should be adjusted again in terms of efficiency.

South Korea established a universal national health insurance (NHI) system covering the entire population since 1999, the year of enactment of the National Health Insurance Act [17, 18]. However, after 20 years, people in South Korea are still paying enormous medical costs because the level of insurance benefits has remained very low. From 2005 to 2018, the percentage of public health resources out of the current health expenditure remained between 56.5% and 59.1%, lower than the average of all the countries in the Organization for Economic Co-operation and Development (OECD; 71~73%) [1].

The South Korean government has tried to raise the benefit rate by implementing several policies, such as the medical aid program and benefit enhancement plan for the four major diseases (FMD) in recent decades [17, 19]. However, almost all the studies using the incidence of CHE reported that there had been no significant impact of the FMD policies [19–21]. These results are rather questionable. First, according to the Ministry of Health and Welfare of Korea [22], the expenditure of NHI has risen six times from 8.8 billion dollars in 2000 to 54.9 billion dollars in 2018. This is a significant increase, even considering the increase in medical use due to aging and the increase in cost due to the development of new medical technology. Nevertheless, the fact that the benefit rate is so stagnant raises doubts that health insurance is functioning inefficiently and that the CHE indicator, especially incidence of CHE, does not assess it properly [22]. Moreover, the FMD program has increased benefit rates significantly on the FMD (all types of cancer, cardiovascular diseases, cerebrovascular diseases, and rare/

intractable diseases) [22]. The benefit rates were calculated by the percentage of health insurance benefits in the total cost of care, including uncovered services. Therefore, there was a discrepancy between the results using benefit rates and studies using the incidence of CHE. Second, most of the studies that analyzed the incidence of CHE reported that medical aid beneficiary households were less likely to face CHE than households enrolled in NHI in South Korea [23–25]. However, these results presented only relative comparisons and did not confirm how much the medical aid program actually reduces CHE.

The new analysis method we develop can analyze the effect of the national health insurance benefits coverage (NHIBC; $TS_{cat}$) on reducing CHE. Therefore, this study has the following objectives. First, we determine how much the NHIBC reduces the incidence and intensity of CHE for all households using the new method. The new method requires a unique dataset, which includes key variables, income, OOP, and the total health expenses (OOP + health insurance benefits) at the individual level. We use the Korea Health Panel Survey (KHPS) data as they include these variables. Second, we utilize the concentration index of NHIBC to confirm on which income class the NHIBC concentrates. Third, we plot graphs for the distribution of CHE and NHIBC to identify how much of the insurance payments are used to prevent CHE and how it should be rearranged in using the finances of NHI. In addition, we infer the level of coverage by separating households with NHI and those with medical aid programs. Fourth, we apply a panel two-part model to analyze the incidence and intensity together. Most previous studies conducted logistic regression analysis when analyzing CHE [26–29], focusing only on the aspect of incidence. However, we analyze the effects on the CHE's incidence and intensity using the panel two-part model (Model 1) and introduce the NHIBC variable ($TS_{cat}$) to Model 1 to determine how health insurance has affected factors related to the CHE. Fifth, we compare the characteristics of the existing CHE incidence and intensity indicators and the newly developed methods ($SH_{cat}$, $TS_{cat}$). For this purpose, the traditional CHE is analyzed and presented in all analyses together including the descriptive statistics, concentration index, graph, and two-part model in this study.

## Methods

### Traditional method for measuring CHE

The traditional method for measuring CHE is to set household income (HI) as the denominator and OOP as the numerator. When the expenditure exceeds a certain threshold, Z, it is considered "catastrophic," as presented in Eq (1) [4]:

$$\frac{OOP}{HI} > Z, \tag{1}$$

Fig 1 is similar to the one used by Wagstaff and van Doorslaer [3] to make CHE easier to understand. The horizontal axis represents the cumulative share of the sample, ordered according to the ratio of medical costs/HI, beginning with individuals with the largest ratio, while the vertical axis represents medical payment as a share of income. In this section, we focus on the lower curve, labeled as OOP in Fig 1, and not the upper one. The lower curve represents the OOP/HI (OOP curve), and $Z_{cat}$ indicates the thresholds. The incidence rate of CHE (headcount of CHE) is equal to

$$H_{cat} = \frac{1}{N} \sum_{i=1}^{N} E_i, \tag{2}$$

where $N$ is the sample size, and $E_i$ is a binary variable, which is 1 if OOP/HI is greater than or equal to $Z_{cat}$, and 0 otherwise. $H_{cat}$ represents the headcount of the CHE, which alone does not

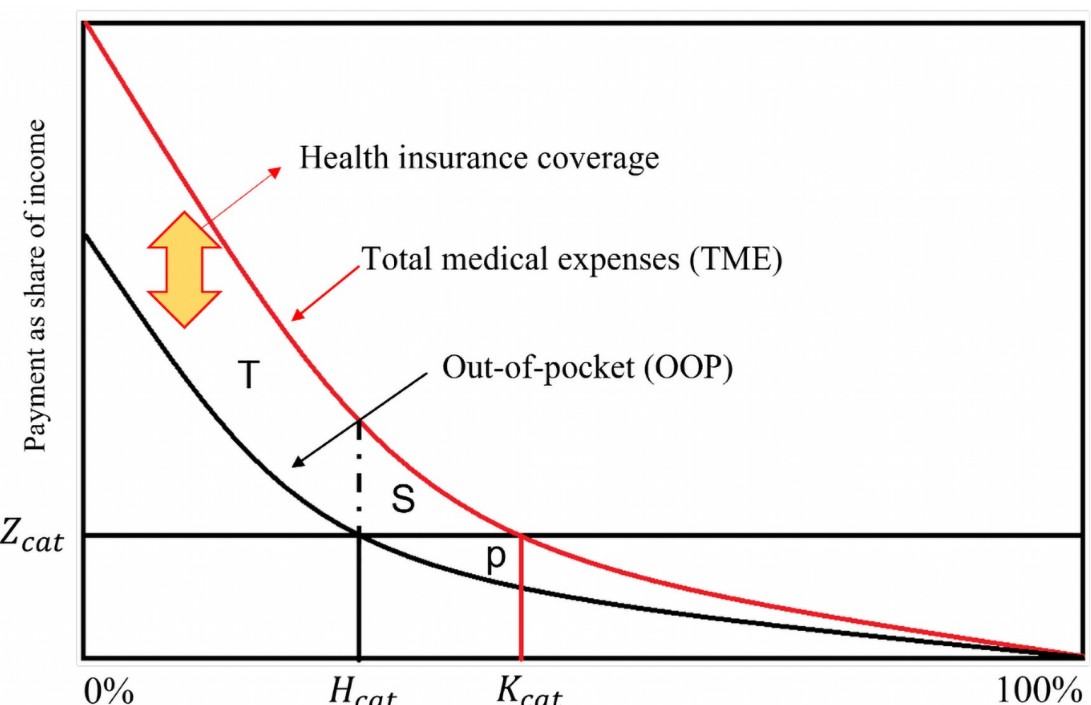

**Fig 1. Two types of medical expenditures as the share of income, by cumulative percentage of population.** The horizontal axis represents the cumulative share of the sample, ordered according to the ratio of OOP to the household income (HI), beginning with individuals with the largest ratio, while the vertical axis represents medical payment as a share of income. There are two curves, with the upper one being the TME/HI (TME curve), and the lower one representing the OOP/HI (OOP curve). $Z_{cat}$ is the threshold, $H_{cat}$ is the incidence of CHE based on OOP/HI, and $K_{cat}$ is based on TME/HI.

indicate to what extent OOP/HI exceeds $Z_{cat}$. Therefore, to know the intensity rate of CHE, we use the following equation:

$$G_{cat} = \frac{1}{N} \sum_{i=1}^{N} O_i, \qquad (3)$$

where $G_{cat}$ is the "catastrophic payment gap" [3]. $O_i$ is calculated by OOP/HI-$Z_{cat}$, which indicates to what extent the OOP payment (as a proportion of income) exceeds the threshold, $Z_{cat}$. However, since this measures the gap based on the total population, it cannot consider to what extent $Z_{cat}$ is exceeded for those who incur CHE ($O_i$ might be added as 0 for those who do not incur CHE). Therefore, we calculate $MPG_{cat}$ to estimate the extent to which $Z_{cat}$ is exceeded for those who incur CHE. $MPG_{cat}$ is called the "mean positive gap" [3] and can be generated by dividing $G_{cat}$ by the number of households that have CHE, as in the following equation:

$$MPG_{cat} = \sum_{i=1}^{N} O_i / \sum_{i=1}^{N} E_i. \qquad (4)$$

Another subject that has to be considered in calculating CHE is inequality. Most studies on CHE mainly focused on the incidence of CHE, whereas they were less concerned about the distributional aspects until recently. However, the distributional aspects are essential because most societies care more about the risk for the poor [3]. Moreover, originally, the CHE indicator was developed to achieve the WHO's goal of "fairness in financial contribution." The

inequality can be analyzed by using the concentration index, so that we apply it to CHE. To calculate the concentration index for $E_i$, we define it as $C_E$ [30].

## Estimating health insurance coverage using the CHE calculation

This subsection clarifies how to estimate the level of health insurance coverage using the CHE measurement method. By substituting the total medical expenses (TME) for OOP, we have

$$\frac{TME}{HI} > Z. \tag{5}$$

HI is the denominator of Eqs (1) and (5), with the only difference being the medical cost. TM−OOP indicates the health insurance benefits (HIB); however, we do not use HIB as the numerator, since, for example, if we set HIB/HI, and Z, to 10%, it would be interpreted as having received HIB equal to more than 10% of the income. Thus, it cannot show the amount by which health insurance covers the CHE.

In Fig 1, there are two curves, with the upper one being the TME/HI (TME curve), and the lower one representing the OOP/HI (OOP curve). The part of the curve that is higher than $Z_{cat}$ is defined as CHE. Since OOP is part of the TME, the OOP curve lies below the TME curve automatically. Eqs (6–8) are similar to Eqs (2–4), with only one difference, which is that they use TME instead of OOP. In Eq (6), $T_i$ is 1 when TME/HI is higher than $Z_{cat}$, and 0 otherwise. Thus, $K_{cat}$ is the incidence rate of CHE based on the TME. In Eq (7), $J_{cat}$ is similar to $G_{cat}$ and $U_i$ is TME/HI−$Z_{cat}$. Moreover, $MPJ_{cat}$ is identical to $MPG_{cat}$.

$$K_{cat} = \frac{1}{N}\sum_{i=1}^{N} T_i \tag{6}$$

$$J_{cat} = \frac{1}{N}\sum_{i=1}^{N} U_i \tag{7}$$

$$MPJ_{cat} = \sum_{i=1}^{N} U_i \Big/ \sum_{i=1}^{N} T_i \tag{8}$$

Therefore, we propose new approaches for measuring the coverage degree of health insurance, that is, estimating how insurance can reduce CHE, using the incidence and intensity rates of CHE. First, using the incidence rate of CHE, in Fig 1, when two curves meet at $Z_{cat}$, it goes to $H_{cat}$ and $K_{cat}$. $H_{cat}$ represents the headcount of the CHE based on OOP, which indicates the incidence rate of CHE, which is reduced by the health insurance coverage. Meanwhile, $K_{cat}$ represents the incidence of CHE without the health insurance coverage. Therefore, $K_{cat}$−$H_{cat}$ (i.e., $SH_{cat}$) indicates households that are exempt from incurring CHE through the health insurance coverage. $SH_{cat}$ represents the extent to which health insurance lowers the incidence rate of CHE, as follows:

$$SH_{cat} = K_{cat} - H_{cat}. \tag{9}$$

Second, using the gap of CHE, in Fig 1, the height difference between the two curves is the difference between TME and OOP, HIB (since both share identical HI, it does not count). However, as mentioned earlier, we do not use $HIP/HI$−$Z_{cat}$. The question we are concerned with is, "To what extent is an individual covered when health care costs account for more than 10% (when the threshold set at 10%) of the household income?" Accordingly, we obtain the health insurance coverage using $J_{cat}$−$G_{cat}$. However, if we just calculate $J_{cat}$−$G_{cat}$, the area of P, under the line of $Z_{cat}$, could be miscalculated. Since $J_{cat}$ is the sum of the heights of the TME curve between 0 and $K_{cat}$ on the x-axis and $G_{cat}$ is the sum of the heights of the OOP curve

between 0 and $H_{cat}$, when we calculate $J_{cat} - G_{cat}$, the area of P will be subtracted together. Therefore, to solve this problem, we need to divide the areas of T and S, find the average gap separately, and then combine the two (we call this combination $TS_{cat}$).

$$Area \ of \ T = J_{cat} - G_{cat} \ when \ E_i = 1, \ T_i = 0$$

$$Area \ of \ S = J_{cat} - G_{cat} \ when \ E_i = 0, \ T_i = 1$$

$$TS_{cat} = T + S$$

Consequently, $TS_{cat}$ is the coverage degree of health insurance when $TME/HI > Z_{cat}$. Potentially, CHE is incurred in the condition of no insurance.

## Data source and study population

We used data from the KHPS version 1.6 (2011–2017) database, which is jointly established by the National Health Insurance Service and the Korea Institute for Health and Social Affairs. The KHPS collects data annually for analysis of medical usage patterns and medical expenditures in South Korea. KHPS data are considered representative of the whole population as the survey employs two-stage stratified random cluster sampling based on the Population and Housing Census. In particular, the KHPS captures medical expenses in detail by dividing them into NHI benefits, statutory payments, and payments for uncovered services, the subject of our study. Furthermore, it is considered a reliable data source as it prevents loss of information and recall-bias errors through health insurance data and receipt checks at a public institution (i.e., the National Health Insurance Services). We used all samples of KHPS, except observations with missing values of variables, in the analysis. The main reason for missing data was that although OOP data were included in the data set, there were no total medical cost data. This case does not seem correct logically. This case may involve the use of medical institutions or drugs from outside the system. KHPS checks TME through health insurance data and receipts, but non-institutional rights (herbal medicine, health food, etc.) may not be investigated. Since our study deals with an institutional system, NHI, these missing data have been removed. The percentage of missing values in the total sample of all years was about 15%, except for 2017 when it was 12%. The final samples included 4,161, 4,508, 4,511, 5,991, 5,752, 5,675, and 5,644 households from 2011 to 2017, respectively. This study was approved by the Korea University Institutional Review Board for IRB exemption (KUIRB-2020-0026-01). All the analyses were performed using the statistical software program, Stata/SE version 14.0 (Stata Corp., Texas, USA).

## Panel two-part model

The two-part model is used to analyze determinants of variables with a high ratio of zero values in the total population, such as medical expenses [31]. The model consists of two parts. The first part analyzes the effect of the factors on whether or not medical services were used through the logit or probit model. The second part involves performing an ordinary least squares (OLS) regression analysis on the subsample that used medical services. The basic assumption of the two-part model related to medical use is that medical use is primarily determined by personal characteristics such as gender, marital status, and health status; on the other hand, the amount of medical use is more influenced by the type of health insurance and the individual's economic status.

We conducted a panel two-part model analysis based on the assumption that factors that determine the incidence and intensity of CHE are different. At this time, two separate models

were analyzed and compared, the model not including $TS_{cat}$ (Model 1) and the model including it (Model 2), to see how the NHIBC ($TS_{cat}$) affects the factors influencing CHE. The equation of Model 1 consists of the following parts 1 and 2:

Part 1: $\log \left(\frac{P}{1-P}\right)_{it} = \beta_0 + \beta_1 X_{1it} + \beta_2 X_{2it} + \beta_3 X_{3it} + u_i + \epsilon_{it},$

Part 2: $\log \left(Y|y > 0\right)_{it} = \beta_0 + \beta_1 X_{1it} + \beta_2 X_{2it} + \beta_3 X_{3it} + u_i + \epsilon_{it},$

where

P: probability of incidence of CHE (threshold: 10%);

Y: intensity of CHE (threshold: 10%);

$X_{1it}$: predisposing factors at point $t$ (gender, age, educational level, marital status, and occupation type of householder);

$X_{2it}$: needs factors at point $t$ (whether or not disabled, number of chronic diseases, and experience of medical use for the FMD);

$X_{3it}$: enabling factors at point $t$ (income adjusted by household size, with or without private health insurance, and type of NHI);

$u_i$: time-invariant term; and

$\epsilon_{it}$: random error term.

When analyzing the panel two-part model, we first used the Breusch-Pagan Lagrange multiplier test and Hausman test to select a suitable model among the pooled OLS model, fixed-effect model, and random-effect model. As a result of the Breusch-Pagan lagrange multiplier test, the pooled OLS model was not suitable for either model. The Hausman test results showed that the fixed-effect model for Model 1 and the random-effect model for Model 2 were more suitable. However, since this study aims to compare models, we suggest conducting the same random-effect model for both models. The random-effect model assumes no correlation between $u_i$, which represents an individual characteristic that does not change over time, and the independent variable. In other words, there must be a random error between $u_i$ and the subject and no correlation between $u_i$ and the independent variable [32].

The KHPS used in this study satisfies the assumption of the $u_i$ and subjects because the survey target households were extracted as a probability sample. Besides, as the independent variable used in this study is the household characteristics, it is difficult to have a significant correlation with $u_i$ at the individual level. Furthermore, some scholars note that researchers can choose a model from among the fixed-effect or random-effect model according to their research purposes [32]. If the purpose is to compare and infer the effects within the sample, the fixed-effect model is suitable. Otherwise, if the purpose is to induce inference that explains the characteristics of the population as a whole, the random-effects model is appropriate. Therefore, in this study, the random-effects model was applied regardless of the Hausman test result.

**Variables for the panel two-part model.** We used variables of the healthcare utilization model of Andersen and Newman [33] in the panel two-part model (Table 1). The variables include the predisposing factors (gender, age, educational level, marital status, and occupation type of householder), enabling factors (income adjusted by household size, with or without private health insurance, and type of NHI), needs factors (whether or not disabled, number of chronic diseases, and experience of medical use of FMD). When adjusting income by household size, we used the equivalence scale of the WHO [7], (*number of adults* + 0.5 × *number of children*)$^{0.56}$. Also, the FMD included cancers, cerebrovascular diseases, cardiovascular diseases, and rare diseases according to the regulation of the Korean Ministry of Health and Welfare. If a household member used medical care (emergency or outpatient or hospitalization) at least once due to any disease among the FMD, it was classified as a household with FMD.

**Table 1. Independent variables.**

| Variables | | Coding |
|---|---|---|
| Predisposing factors | Gender | 0: Men; 1: Women |
| | Age | 0: <29; 1: 30~39; 2: 40~49; 3: 50~64; 4: >65 |
| | Educational level | 0: Higher than college; 1: High school; 2: Less than middle school |
| | Marital status | 0: Married; 1: Single |
| | Occupation type | 0: Employee; 1: Employer or self-employed; 2: Unemployed |
| Enabling factors | Income level | 0: Richest; 1: Quintile 4; 2: Quintile 3; 3: Quintile 2; 4: Poor |
| | Type of NHI | 0: Civil servant; 1: Employee; 2: Self-employed; 3: Medical aid beneficiary |
| | Private health insurance | 0: No; 1: Yes |
| Needs factors | Disabled | 0: No; 1: Yes |
| | FMD | 0: No; 1: Yes |
| | Chronic diseases | Continuous |

Note: NHI: national health insurance; FMD: four major diseases.

## Results

### General characteristics of the sample

Table 2 shows the general characteristics of the sample. Because of space limitations, the characteristics of only the 2017 sample are presented here. First, in terms of householder characteristics, men were the most common with 76.3%, and 33.4% of the sample was aged over 65 years. High school accounted for 38.7%. Regarding marital status, married was more common than single. As for occupation type, unemployed was the most common (43.1%). This seems to be attributable to the increase in the number of older people as household heads. Second, in terms of household characteristics, 26.5% were the richest and 16.3% were poor. Further, 48.6% of households had private medical insurance. Among the types of NHI, the employee type was the most common at 60.1%. There were 10.8% households with disabilities, and 24.5% of households had FMD. The average amount of chronic diseases was 0.87%.

### Effect of health insurance benefits on reducing the incidence and intensity of CHE

In Table 1, the left-side columns named "Based on OOP payment" present the traditional OOP-based CHE, the middle columns called the "Based on TME" present the newly calculated TME-based CHE values in this study, and the right-side columns called the "National health insurance benefits coverage" present the effect of the NHI on reducing the incidence and intensity of CHE.

First, $H_{cat}$, $G_{cat}$, $K_{cat}$, $J_{cat}$, $TS_{cat}$, and their concentration indexes $C_E$, $KC_E$, $C_o$, $KC_o$, $TSC_o$, respectively, decrease as the threshold value increases. This means that fewer people pay high medical bills, and the people who meet the high threshold are concentrated in the low-income group. On the other hand, all types of "mean positive gaps" ($MPG_{cat}$, $MPJ_{cat}$, $MPTS_{cat}$) increased as the threshold increased. This is because when a higher threshold is set, only those who have a high ratio of medical expenses to income are included in the average calculation. Among the indicators, $SH_{cat}$ showed a slightly different pattern. The $SH_{cat}$ value slightly increased between the threshold of 2.5% and 5% but decreased at 10%. This seems to be due to many cases wherein health insurance covers less than 10% of the TME compared with income.

**Table 2. General characteristics, 2017.**

| Variables | | | N (%) |
|---|---|---|---|
| Characteristics of householders | Gender | Men | 4,305 (76.3) |
| | | Women | 1,339 (23.7) |
| | Age | <29 | 187 (3.3) |
| | | 30~39 | 676 (12.0) |
| | | 40~49 | 1,166 (20.7) |
| | | 50~64 | 1,727 (30.6) |
| | | >65 | 1,888 (33.4) |
| | Education | Higher than college | 1,756 (31.1) |
| | | High school | 2,185 (38.7) |
| | | Less than middle school | 1,703 (30.2) |
| | Marital status | Married | 3,929 (69.6) |
| | | Single | 1,715 (30.4) |
| | Occupation type | Employee | 1,356 (24.0) |
| | | Employer / self-employed | 1,858 (32.9) |
| | | Unemployed | 2,430 (43.1) |
| Characteristics of households | Income level | 5th (Richest) | 1,496 (26.5) |
| | | 4th | 1,190 (21.1) |
| | | 3rd | 1,027 (18.2) |
| | | 2nd | 1,009 (17.9) |
| | | 1st (Poor) | 922 (16.3) |
| | Private health insurance | No | 2,901 (51.4) |
| | | Yes | 2,743 (48.6) |
| | Type of NHI | Civil servant | 344 (6.1) |
| | | Employee | 3,387 (60.1) |
| | | Self-employed | 1,481 (26.2) |
| | | Medical aid | 432 (7.6) |
| | Presence of disabled | No | 5,037 (89.2) |
| | | Yes | 607 (10.8) |
| | Presence of FMD | No | 4,259 (75.5) |
| | | Yes | 1,385 (24.5) |
| No. of chronic diseases (Mean/S.D.) | | | 0.87 (0.87) |
| No. of sample | | | 5,644 |

In other words, it is highly likely that the medical expenses covered by health insurance are concentrated on low-price medical services rather than on high price medical services.

What is interesting is the concentration index. In Fig 2, $C_E$ and $SC_E$ have a large difference according to the threshold, while $C_o$ and $TSC_o$ have a small difference. According to the threshold, the meaning of the deviation of the concentration index can be related to the indicator's reliability, which is explained in the Discussion section.

Fig 3 presents a graph that analyzed the 2011 and 2017 KHPS data in the same manner as in Fig 1. This figure presents the analyzed graph by setting the threshold to 40%, because a setting of less than 40% indicates poor visibility. Here, the bar graph shows the degree of mitigation of the CHE's intensity aided by the NHI benefits ($TS_{cat}$). The reason why OOP/HI appears as a smooth curve and TME/HI as jagged bar graphs is because the order of the x-axis is set in the order of OOP/HI. Specifically, for example, if household A incurred more TME than household B but received more health insurance benefits and paid less OOP, household A would be placed to the left side of household B in order. In the left-side plot of Fig 3, the bar graph

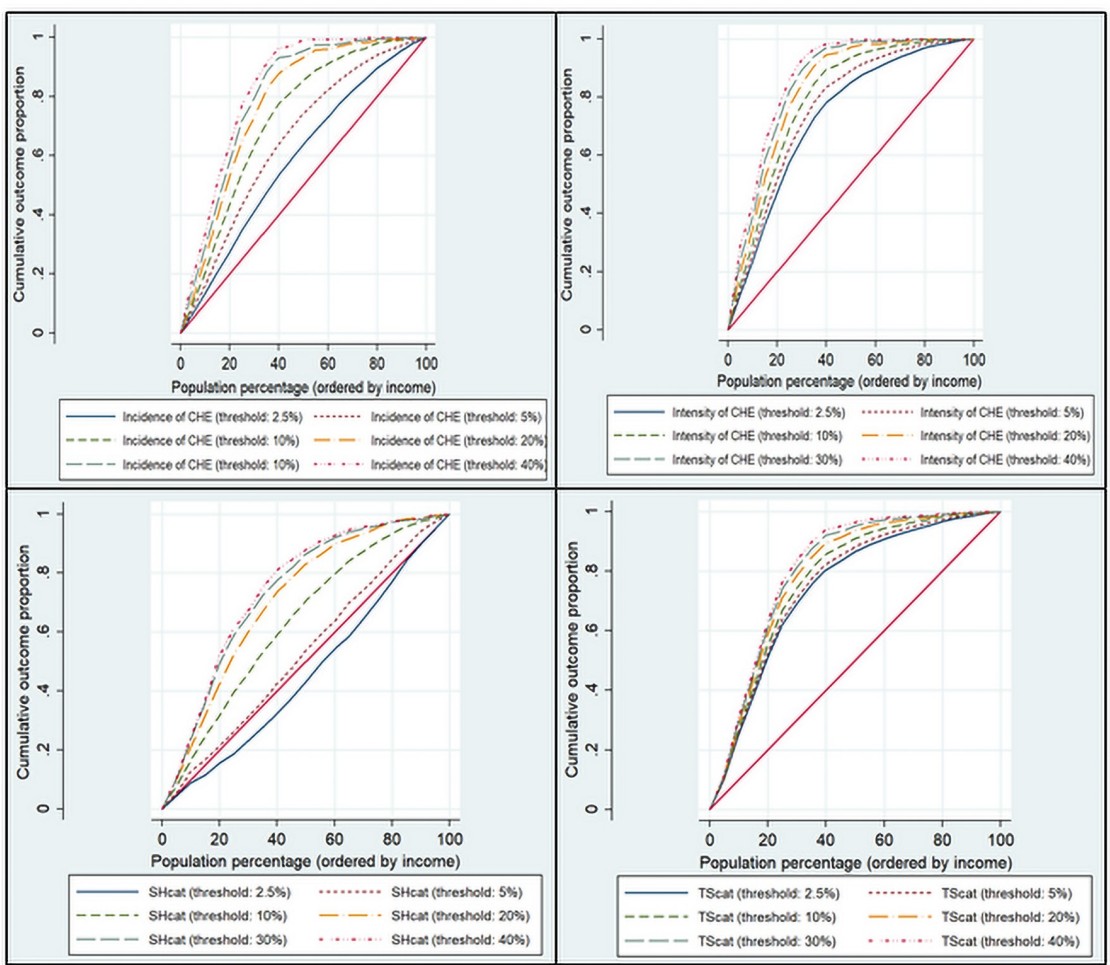

**Fig 2. Concentration curve of indicators, 2017.** The upper left graph is the $C_E$, and the right is the $C_o$; the lower left graph is the $SC_E$, and the right is the $TSC_o$. The red line in each plot is the equality line. The rest curves represent the distribution of the catastrophic health expenditure (CHE) and national health insurance benefits coverage incidence or intensity by household income according to several thresholds. The further is the concentration curve from the equality line, the more it is concentrated on the lower-income class.

represented high in the middle of the x-axis means that despite the high total medical cost incurred, health insurance benefits are very high so that the OOP ranking becomes relatively low.

The higher the bar graph ($TS_{cat}$), the greater the effect of mitigating the intensity of CHE. The $TS_{cat}$ considers the relative ratio of income and medical expenses, and so if the income level is low, the effect of medical benefits becomes greater. The left plot is a graph analyzing the entire sample, and the right excludes the medical aid recipients. We can see that the high bar graph in the middle of the x-axis, shown in the left plot, has disappeared in the right plot. indicating that medical aid recipients are reducing the intensity of CHE by this much.

## Traditional CHE and NHIBC on CHE in a time series

Table 3 shows the trends of traditional CHE indicators ($H_{cat}$, $C_E$, $G_{cat}$, $MPG_{cat}$, $C_o$) and health insurance coverage indicators ($SH_{cat}$, $SC_E$, $TS_{cat}$, $MPTS_{cat}$, $TSC_o$) from 2011 to 2017 (threshold: 10%). First, the incidence of CHE, $H_{cat}$, and its concentration index, $C_E$, were almost

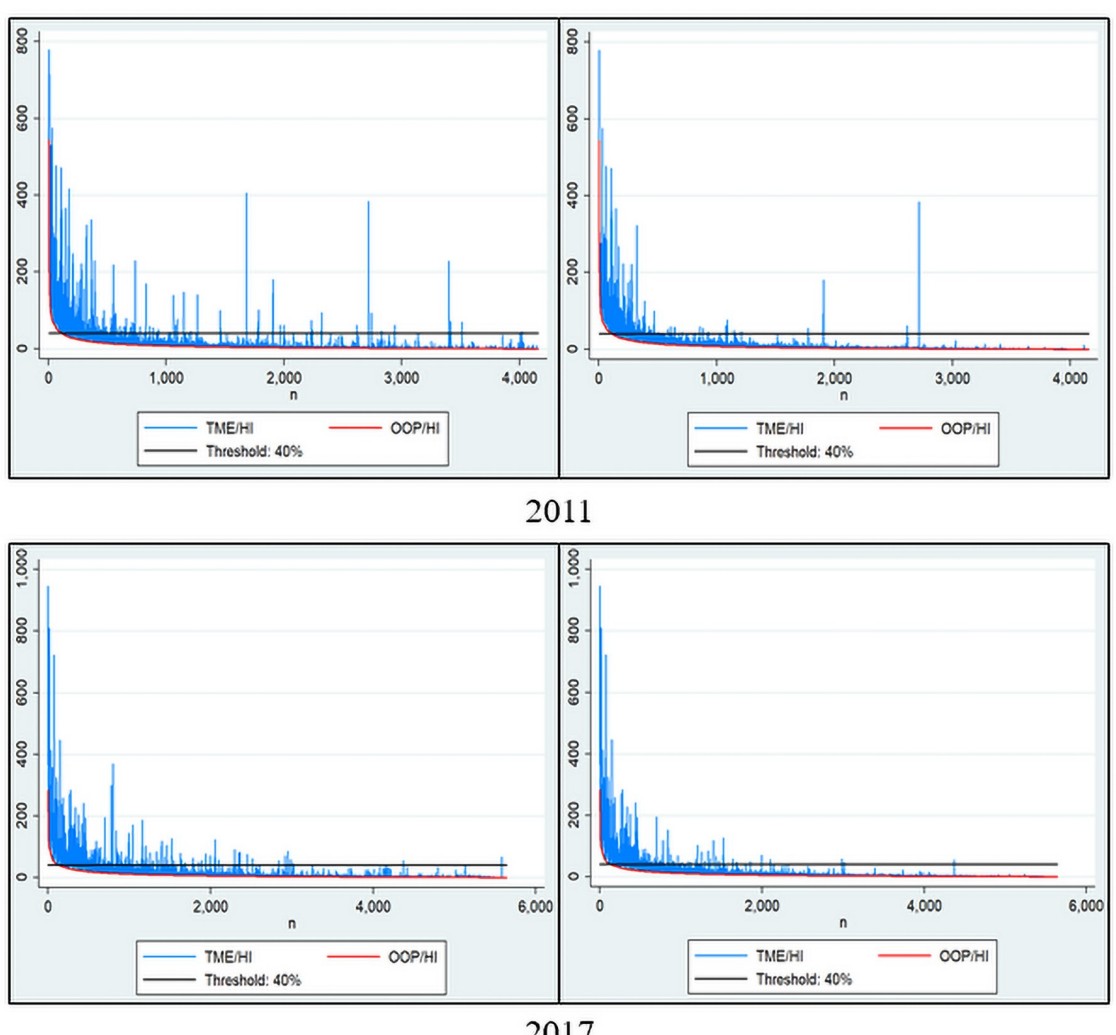

**Fig 3. Health insurance coverage on the incidence and intensity of CHE (2011, 2017).** The graphs represent the how the national health insurance (NHI) mitigates the intensity of the catastrophic health expenditure (CHE). The y-axis is the proportion of out-of-pocket (OOP) expenses or total medical expenses (TME) to income. The x-axis starts with households with the highest proportion of OOP and TME to income and ranks them in descending order. The red curve is the OOP/HI, and the blue bar graph is the difference between the TME/HI and OOP/HI, which is the national health insurance benefits coverage. HI: household income.

unchanged over time; however, $MPG_{cat}$ decreased by about 7% points from 21.38% in 2011 to 14.4% in 2017. Since the 7% reduction may not sufficiently represent the difference between the incidence and the intensity indicator, we recalculated $H_{cat}$, $G_{cat}$, and $MPG_{cat}$ for households with FMDs, which are the target for strengthening the health insurance coverage in South Korea. As a result, the $H_{cat}^*$ of households with FMDs was 37%~39%, which was about 20% higher than that of all households at 17~19% and also hardly changed over time; on the contrary, $MPG_{cat}^*$ decreased by about 12% from 2011 to 2017.

$SH_{cat}$ refers to the percentage of households exempt from CHE through health insurance benefits. There was a slight increase or decrease in time series, but only about 1% of change was observed in all years. Moreover, $SC_E$ is around -0.3 in all years, indicating that this trend has not changed significantly. $TS_{cat}$ continued to decrease from 8.85% in 2011 to 6.65% in 2014, and then it increased to 8.81% in 2016, recovering to the level in 2011. $MPTS_{cat}$ showed a

**Table 3. Traditional CHE and NHI benefit coverage on CHE (2011–2017; threshold: 10%).**

| Indicators | | 2011 | 2012 | 2013 | 2014 | 2015 | 2016 | 2017 |
|---|---|---|---|---|---|---|---|---|
| Traditional catastrophic health expenditure (CHE) indicators | $H_{cat}$ | 17.67% | 17.83% | 17.72% | 17.41% | 18.88% | 19.80% | 19.26% |
| | $C_E$ | -0.492 | -0.472 | -0.492 | -0.501 | -0.510 | -0.508 | -0.525 |
| | $G_{cat}$ | 3.77% | 3.84% | 2.77% | 2.78% | 3.22% | 3.24% | 2.76% |
| | $MPG_{cat}$ | 21.38 | 21.56 | 15.65 | 15.98 | 17.09 | 16.43% | 14.4% |
| | $C_o$ | -0.669 | -0.736 | -0.669 | -0.642 | -0.656 | -0.652 | -0.653 |
| | $H_{cat}^*$ | 38.28% | 39.49% | 38.21% | 37.56% | 39.86% | 38.40% | 37.40% |
| | $G_{cat}^*$ | 10.78 | 9.29 | 7.01 | 7.34 | 7.33 | 6.85 | 5.99 |
| | $MPG_{cat}^*$ | 28.23 | 23.54 | 18.34 | 19.54 | 18.40 | 17.85 | 16.06 |
| National health insurance (NHI) coverage on CHE | $SH_{cat}$ | 13.54% | 14.01% | 14.95% | 14.47% | 14.04% | 14.64% | 15.17% |
| | $SC_E$ | -0.348 | -0.303 | -0.348 | -0.314 | -0.332 | -0.371 | -0.351 |
| | $TS_{cat}$ | 8.85% | 8.80% | 7.41% | 6.65% | 7.73% | 8.81% | 8.06% |
| | $TSC_o$ | -0.634 | -0.665 | -0.634 | -0.599 | -0.604 | -0.585 | -0.614 |
| | $MPTS_{cat}$ | 28.43% | 27.73% | 22.72% | 20.91% | 23.54% | 25.62% | 23.47% |

Note 1: Number of entire household samples from 2011 to 2017: 4,165, 4,510, 4,514, 5,995, 5,751, 5,678, and 5,644, respectively;

Note 2:

*: Recalculated for the households with the four major diseases; number of households with the four major diseases from 2011 to 2017: 930, 947, 997, 1,294, 1,302, 1,341, and 1,384, respectively.

trend similar to $TS_{cat}$. $TSC_o$ had a negative value in all years, indicating that $TS_{cat}$ was concentrated in the low-income class.

### Effect of national health coverage on the factors of CHE

In this subsection, we analyze and compare Model 1, which analyzes the effect on incidence and intensity of CHE through a panel two-part model, and Model 2, which adds the NHIBC ($TS_{cat}$) to Model 1.

First, as a result of the Hausman test, the chi-square value of Model 1 was 0.001, and so it was concluded that the fixed-effect model was appropriate. On the contrary, the chi-square value of Model 2 was 0.966, indicating that the random-effect model was suitable. However, since this study aimed to infer the characteristics of the entire population and compare the results of both Models 1 and 2, we used the random-effects model in all models.

The results of Model 1 are presented in Table 4. First, in the panel logistic regression analysis for the incidence of CHE, the probability of incidence of CHE (threshold: 10%) was affected by several characteristics: gender, age, education level, and occupation type among the predisposing factors of householders. The need factors were whether or not households with FMD and the number of chronic diseases. In terms of enabling factors, income level, private insurance, and medical aid recipients significantly affected the CHE.

The panel regression, which analyzed the effect on the intensity of CHE (threshold: 10%), showed a different pattern from the logistic regression. The householder's gender, education level, private insurance, and medical aid did not significantly affect the intensity of CHE, and the effect of the number of chronic diseases was reversely negative. These results can be interpreted as meaning that the householders' gender, education level, private insurance, and medical aid affect the incidence of CHE, that is, up to 10% of medical expenses compared to income, but not beyond that. Moreover, in particular, the result of the number of chronic diseases could indicate that a certain number of chronic illnesses increased the likelihood of

**Table 4. Factors associated with the incidence and intensity of CHE.**

| | | Model 1 | | | | Model 2 | | | |
|---|---|---|---|---|---|---|---|---|---|
| | | Incidence | | Intensity | | Incidence | | Intensity | |
| | | O.R. | S.E. | β | S.E. | O.R. | S.E. | β | S.E. |
| $TS_{cat}$ | | - | - | - | - | 8.969*** | 0.348 | 0.514*** | 0.011 |
| Gender (Men) | Women | 0.688*** | 0.045 | -0.024 | 0.037 | 0.902 | 0.081 | -0.019 | 0.035 |
| Age (<29 years) | 30~39 | 0.797 | 0.178 | 0.127 | 0.178 | 0.556* | 0.159 | -0.073 | 0.159 |
| | 40~49 | 0.762 | 0.164 | 0.220 | 0.171 | 0.648 | 0.178 | -0.043 | 0.152 |
| | 50~64 | 1.301 | 0.277 | 0.343* | 0.167 | 1.019 | 0.277 | 0.145 | 0.149 |
| | >65 | 2.187*** | 0.472 | 0.350* | 0.167 | 0.953 | 0.264 | 0.066 | 0.149 |
| Education level | High school | 1.291*** | 0.094 | 0.050 | 0.045 | 0.816* | 0.081 | -0.054 | 0.041 |
| (Higher than college) | Less than Middle school | 0.962 | 0.06 | -0.023 | 0.046 | 0.927 | 0.081 | -0.09* | 0.041 |
| Marital status (Married) | Single | 1.004 | 0.047 | -0.011 | 0.033 | 1.018 | 0.084 | 0.013 | 0.028 |
| Occupation type (Employee) | Employer/ Self-employed | 0.844*** | 0.042 | -0.113** | 0.034 | 0.854* | 0.062 | 0.073** | 0.029 |
| | Unemployed | 0.826*** | 0.045 | -0.110** | 0.035 | 0.886 | 0.068 | 0.054 | 0.031 |
| Income level (Richest)* | Quintile 4 | 3.321*** | 0.351 | 0.209* | 0.098 | 2.541*** | 0.361 | 0.201* | 0.085 |
| | Quintile 3 | 7.485*** | 0.772 | 0.315** | 0.092 | 4.225*** | 0.586 | 0.219** | 0.081 |
| | Quintile 2 | 16.235*** | 1.695 | 0.617*** | 0.091 | 5.483*** | 0.773 | 0.410*** | 0.080 |
| | Quintile 1 | 41.306*** | 4.571 | 0.930*** | 0.092 | 6.735*** | 1.006 | 0.591*** | 0.082 |
| Private health insurance (No) | Yes | 1.482*** | 0.077 | 0.051 | 0.035 | 1.324*** | 0.096 | 0.071* | 0.032 |
| Type of NHI (Civil servant) | Employee | 1.019 | 0.093 | -0.051 | 0.054 | 1.092 | 0.140 | -0.053 | 0.049 |
| | Self-employed | 0.911 | 0.089 | -0.032 | 0.058 | 0.956 | 0.130 | -0.029 | 0.053 |
| | Medical aid beneficiaries | 0.286* | 0.034 | -0.080 | 0.071 | 0.045*** | 0.007 | -0.391*** | 0.066 |
| Disabled (No) | Yes | 1.100 | 0.077 | -0.001 | 0.038 | 0.784 | 0.075 | -0.065 | 0.036 |
| FMD (No) | Yes | 4.052*** | 0.200 | 0.252*** | 0.028 | 1.348*** | 0.092 | 0.029 | 0.026 |
| No. of chronic diseases | | 1.191*** | 0.059 | -0.078* | 0.034 | 1.104 | 0.077 | -0.077* | 0.03 |
| Constant | | 0.006*** | 0.001 | 1.214*** | 0.195 | 0.002*** | 0.001 | 0.399* | 0.174 |
| No. of samples | | 36,255 | | 6,676 | | 36,255 | | 6,676 | |
| No. of groups | | 7,878 | | 3,029 | | 7,878 | | 3,029 | |
| $R^2$ | | Waldx2 = 3,508.6*** | | Within | 0.044 | Waldx2 = 3,691.99*** | | Within | 0.354 |
| | | -2logL = -12,518.81 | | Between | 0.137 | -2logL = -6,767.48 | | Between | 0.268 |
| | | | | Overall | 0.092 | | | Overall | 0.314 |
| Rho | | 1,229.88*** | | 0.066 | | 545.23*** | | 0.202 | |

Note:

*p < .05;

**p < .01;

***p < .001.

The income level is adjusted by the equivalence scale of the World Health Organization [7]. CHE: catastrophic health expenditure; O.R.: odds ratio; S.E.: standard error; NHI: national health insurance; FMD: four major diseases.

incidence of CHE; however, when the number exceeded a certain level, patients could give up the treatment due to expensive medical costs.

The analysis result of Model 2, which controlled the level of NHIBC ($TS_{cat}$), was slightly different from Model 1. The results are given in Table 4. Age of 30~39 years, high school graduates, employers or self-employed, income level, with private insurance, medical aid recipients, and households with FMD were more likely to affect the incidence of CHE. Samples aged over 65 years, unemployed, and number of chronic diseases significantly affected Model 1 but were not significant in Model 2. These results can be attributed to the fact that

**Table 5. Incidence and intensity of CHE based on OOP and TME in South Korea, 2017.**

| Based on OOP payments | | | | Based on TME | | | | National health insurance benefits coverage | | | |
|---|---|---|---|---|---|---|---|---|---|---|---|
| **Threshold** | **2.5%** | **5%** | **10%** | **20%** | **Threshold** | **2.5%** | **5%** | **10%** | **20%** | **Threshold** | **2.5%** | **5%** | **10%** | **20%** |
| Incidence measures | | | | | | | | | | | | | | |
| $H_{cat}$ | 55.69% | 36.41% | 19.26% | 8.03% | $K_{cat}$ | 71.21% | 53.35% | 34.43% | 19.95% | $SH_{cat}$ | 15.52% | 16.94% | 15.17% | 11.92% |
| $C_E$ | -0.268 | -0.389 | -0.525 | -0.620 | $KC_E$ | -0.213 | -0.312 | -0.448 | -0.548 | $SC_E$ | 0.014 | -0.149 | -0.351 | -0.495 |
| Intensity measures | | | | | | | | | | | | | | |
| $G_{cat}$ | 5.19% | 4.08% | 2.76% | 1.52% | $J_{cat}$ | 14.50% | 12.97% | 10.83% | 8.24% | $TS_{cat}$ | 9.30% | 8.89% | 8.06% | 6.71% |
| $MPG_{cat}$ | 9.33% | 11.21% | 14.4% | 19.14% | $MPJ_{cat}$ | 20.37% | 24.33% | 31.51% | 41.39% | $MPTS_{cat}$ | 13.08% | 16.68% | 23.47% | 33.72% |
| $C_o$ | -0.534 | -0.795 | -0.653 | -0.716 | $KC_o$ | -0.549 | -0.583 | -0.624 | -0.663 | $TSC_o$ | -0.558 | -0.581 | -0.614 | -0.651 |

Note: CHE: catastrophic health expenditure; OOP: out of pocket; TME: total medical expenses.

the level of health insurance benefits ($TS_{cat}$) affected these predisposition factors and chronic diseases.

In a panel regression that analyzed the effect on the intensity of CHE (threshold: 10%), the intensity increased statistically significantly for employers or the self-employed, private health insurance, and lower-income level; however, it decreased in householders with less than middle school education, medical aid recipients, and higher number of chronic diseases. More specific interpretations are presented in the Discussion.

## Discussion

This study attempted to analyze how effectively health insurance benefits reduce CHE in South Korea. First, as a result of a threshold of 10% (Table 5), the benefits of NHI reduced the incidence of CHE by 15.17% and decreased the mean positive gap by 23.47% in 2017. To confirm the reliability Sof this result, we presented the result of calculating the existing traditional CHE together. In this study, the incidence of CHE was found to be 19.26% at the threshold of 10%. This was similar to the results of studies using the same data source and the same calculation method [24, 34–36]. However, the results of the intensity and concentration index of CHE could not be compared because few similar studies were conducted in South Korea.

As one analysis result of intensity, $MPTS_{cat}$ decreased from 28.43% in 2011 to 20.91% in 2014, and it recovered to 25.62% in 2016 (Table 3). The evidence supporting this is as follows. The 2011–2014 period was the second version policy period of the Korean Health Insurance Coverage Enhancement Plan, and 2015–2018 was the third version period. The coverage effect gradually declined during the second version policy period, because medical institutions increased medical expenses again through strategies such as expanding non-benefit medical services in response to government policies [37, 38]. The increase in the early third version period (2015~2016) could be interpreted as the effect of introducing a new policy. Moreover, $TSC_o$ was negative in all years, indicating that the NHI coverage was concentrated in the low-income class (Table 3).

Next, we performed two analyses through the panel two-part model (Table 4). First, we analyzed the factors affecting CHE incidence and intensity (Model 1), and, second, we added $TS_{cat}$ as a control variable to Model 1 and analyzed it (Model 2). The reason for this separate analysis of Models 1 and 2 is to see how the determinant factors change when NHIBC ($TS_{cat}$) on CHE is added to the existing model. As a result, first, it was found that Model 2 had better model fit ($R^2$) than Model 1. Second, the higher the intensity of CHE, the higher the $TS_{cat}$ (Table 4). This is because the South Korean health insurance system applies a fixed-rate payment system in general, and so the health insurance benefit and OOP have a positive correlation. At this time,

since the dependent variable, the intensity of CHE, and the independent variable, $TS_{cat}$, share the same denominator (total household income), the regression coefficient can be interpreted as the size of medical expenses.

The determinant factors of CHE were different in Models 1 and 2. First, gender, age over 65 years, unemployed, and number of chronic diseases were significant in the logistic regression of Model 1 but not of Model 2 (Table 4). The reason for these results may be that health insurance benefits offset these factors. Moreover, in the panel regression analysis, the influencing factors were similar in both Models 1 and 2, except for medical aid and FMD. Medical aid decreased the intensity of CHE in Model 2, but it was not significant in Model 1. This result also shows that Model 2 is a better statistical model. This evidence is supported by the disappearance of the middle x-axis bar graph when households with medical aid recipients are excluded in Fig 3. FMD increased the intensity of CHE in Model 1, but it was not significant in Model 2. This result shows the effect of the South Korean health insurance policy on FMD, which can also be presented as a basis for the better model fit of Model 2 compared with Model 1. This is also supported by evidence from the results in Table 3 showing that $MPG_{cat}$ of the FMDs decreased with time. Therefore, contrary to the results of previous studies that analyzed only the incidence of CHE [19–22], it can be interpreted that the government's policies on FMD are effective.

Meanwhile, the purpose of the health insurance system is to prevent CHE. From this point of view, health insurance benefits should offset CHE. However, it can be seen that the South Korean health insurance has not yet reached that level. Then, should South Korea increase the level of health insurance benefits? Is that the right solution? For this, a premium increase is inevitable, which would become a burden on the people again. If so, what strategies should the South Korean NHI plan to promote in the future? Figs 1 and 3 can provide a clue to answer this question. These figures make it possible to determine to whom health insurance benefits are being paid, whether the effects sufficiently mitigate CHE's intensity, and which groups do not mitigate the intensity.

The policy idea we propose here is to flatten the OOP curve. There are two groups in Fig 3: groups that are higher and lower than the line of $Z_{cat}$. We can imagine that if health insurance reduces the benefit for the lower group and covers the higher group's medical expenses with the financial resources secured through it, there would be no need to increase the premium. This theoretical explanation can be converted into a policy explanation as follows. Health insurance increases the OOP rate for low-cost medical services such as cold treatment; it then uses the finances secured through this to expand the coverage of high-cost medical services. Since the prevalence of mild diseases is generally higher than that of severe diseases, even a slight increase in the OOP rate for mild illnesses could secure sufficient resources to cover medical expenses for severe diseases. Similarly, another strategy might be for the health insurance to raise the OOP maximum for the high-income class and use the finances secured through this to lower the OOP maximum for low-income groups.

Public resistance to these proposals may be a concern. Still, this method would be easier than raising health insurance premiums. Health insurance premiums are compulsory tax paid regularly and permanently even if one does not use medical care. Moreover, medical expenses are temporary, and most South Korean people admit the responsibility for medical costs. Then, how should the health policy be set in detail? This can be inferred from the results of the independent variables in Model 2 (Table 4). The panel regression analysis of Model 2 found that some predisposing factors (less than middle school graduates, employers, and self-employed persons) had an effect. Still, their regression coefficients were low, and the p-value was also significant within 0.1%. Therefore, it was difficult to see that if they had a great influence on the intensity of CHE. Instead, the effects of income level and medical aid on the

intensity of CHE were still high. These results show that South Korea's health insurance policy is paying a lot to medical aid recipients, and the coverage for the low-income class of health insurance is relatively insufficient. In addition, the results showed that FMD's effect on the intensity of CHE is statistically significant in Model 1, but not in Model 2. This means that the benefit enhancement plan for the FMD was effective. However, the impact of the income level on the intensity of CHE remains. This suggests that the coverage effect was inefficient. Therefore, it is necessary to modify the policy of strengthening South Korea's health insurance coverage from focusing on specific diseases (FMD) to considering income level (OOP maximum system), and it is necessary to reorganize the current medical aid system, which causes a financial burden for health insurance.

To further expand the implications of this study, we discuss the limitations of the existing CHE indicators noticed through the supplemental analysis. First, we consider that using only the incidence of the CHE indicator in measuring the effectiveness of health policy is a limited method. In fact, many studies usually apply logistic regression models when analyzing CHE because the demographic factors' effects are well represented and statistically significant. The determinants of CHE incidence appear different for each study because the data sources and designs used are different, although many variables are reported to have an influence. For example, Lee and Shin [39] reported that gender, education level, employment status, marital status, number of chronic diseases, and presence of the elderly aged over 65 influenced CHE incidence. Lee et al. [36] confirmed that the householders' age; type of NHI; income level; and type and number of chronic diseases including cerebral diseases, renal failure, and neoplasm influenced CHE incidence. Kim and Sakong [35] concluded that age, marital status of the householder, type of NHI, income level, number of household members, and number of chronic diseases in the household influenced CHE incidence. Especially, the determinant variables suggested by Kim and Sakong [35] substantially overlap with our results of Model 1. This may be because they used the same KHPS data and applied the same panel logistic regression analysis as in this study.

However, the fact that the determinants appear differently for each study may increase the reliability of the results. In fact, the logistic analysis method for the incidence of CHE has the advantage of being able to determine which groups are vulnerable among the entire population. Moreover, since there are many statistically significant variables, it is useful to make policy suggestions through them. However, if we were interpret it in reverse, the incidence approach is an indicator that is greatly influenced by demographic characteristics. For example, older people are highly likely affected by diseases so that they could spend more medical expenses. Eventually, there is high possibility that the incidence of CHE will increase. This approach allows policy recommendations such as preventive health care to prevent CHE from occurring. However, if the strategy for strengthening health insurance coverage is to reduce OOP, a decrease in OOP will appear in people who incur medical expenses. Therefore, considering intensity rather than incidence would be more appropriate in this case.

In Fig 2, the concentration curves are represented for each of several thresholds. $C_E$ and $SC_E$, corresponding to the incidence approach, had a relatively larger difference according to the thresholds compared with the intensity approach, $C_o$ and $TSC_o$. This means that the incidence approach is more affected by the threshold, and intensity is relatively less affected by the threshold. In addition, $SC_E$ showed a slightly positive value at a threshold of 2.5%. This result means that the high- or middle-income group is more sensitive to the low threshold, and it is highly likely that the middle-income group would be exempt from CHE first when the OOP decreases. Further, even if there is a high OOP to income ratio—for example, 50% becomes 12%, which is a significant decrease—there is no perceived change from the point of view of the incidence approach. The incidence of CHE cannot provide insights into these points.

Evidence that intensity has more advantages over incidence can also be seen in Table 3. $SH_{cat}$ showed little fluctuation over time. Moreover, the $H_{cat}$ of all households and of households with FMD hardly changed over time. On the contrary, $MPG_{cat}$ decreased significantly from 2011 to 2017. In fact, almost all studies that performed difference-in-differences analysis on policy effects by comparing the CHE incidence of the policy-beneficiary group and the non-beneficiary group concluded that there was no effect [19–21, 40, 41]. To sum up, these results represent that the incidence approach is insensitive and intensity is more sensitive; however, there are few studies that used intensity.

The threshold of CHE has already been criticized for being ad hoc in previous studies [14]. Additionally, if the difference in the result value according to the threshold is large, reliability as an index could be degraded. Besides, since the incidence is measured in a dichotomy, there is a problem that the determination of whether or not it occurs varies depending on the threshold. For example, if household A has a 15% ratio of medical expenses to income and a policymaker sets the threshold to 10%, A will be classified as a CHE incurring household. Moreover, if the threshold is set to 20%, household A would be classified differently as a non-occurring household. On the other hand, the intensity approach may be more suitable for determining policy priorities because even if thresholds are set differently, households with a high value of CHE intensity can be considered as those with the highest economic burden.

For the abovementioned reasons, we presented $TS_{cat}$ as a more important result than $SH_{cat}$. This is why we used $TS_{cat}$ in the panel two-part model rather than $SH_{cat}$. The results of the Hausman test, which was conducted before the panel two-part model, can also show the usefulness of our new variables ($TS_{cat}$). As we figured out, Model 1 showed that error term $u$, which does not change over time, is correlated with the independent variable, so that the fixed-effect model was appropriate, whereas Model 2 showed that the random-effect model was suitable because it was not correlated. If error term $u$ is correlated with the independent variable, there may be some bias in the sample or some independent variables may not have been considered. Further, it is generally known that when the assumptions are satisfied, the efficiency of estimation for random-effects is higher than that for the fixed-effect model [32]. Even when looking at $R^2$, there was no significant difference in the logistic model; however, in the OLS panel regression analysis, Model 2 represented a better model fit (Table 4).

This study is meaningful in that it took a different approach from the existing CHE studies, but it contains limitations. First, it is difficult to apply the new proposed measurement method to other countries because it is impossible without household unit data on the OOP, TME, health insurance benefits, income, and living expenses. Second, it has limited understanding of the effects of private insurance. Although private insurance was included and analyzed in the panel two-part model, it was impossible to present a graph or basic statistics on how much private insurance reduces CHE. This is because private insurance is enrolled on an individual basis rather than as a family unit. Third, traditional CHE calculation measures may not fully represent the economic status in developed countries because they simply define the household's ability to pay as income. In high-income countries, there are many households with incomes higher than the cost of living, creating lots of assets. Accumulated assets strengthen the ability to pay. However, we could not consider the effects of these assets in the study. This is because not only does the KHPS not provide data on assets or liabilities, but also there are not many methods to analyze this. Future studies should research this issue further.

This study analyzed the intensity of CHE, which has not been attempted much in South Korea and explained clear differences from the incidence approach. In addition, this study is the first to introduce a two-part model in the framework of the existing studies, which mainly attempted logistic regression analysis focusing on only the incidence; therefore, this study is meaningful in terms of its methodology. Especially, this study presented a new idea for the

health insurance policy direction by combining the existing CHE indicator and a new method modified from this indicator. The South Korean NHI focuses on specific diseases and strengthens its coverage, but this method has limitations in efficiently reducing CHE. Therefore, according to the several analyses using the new method, we concluded that health insurance should pay for benefits based on the income level and reorganize the medical aid system.

## Acknowledgments

We would like to thank Editage (www.editage.co.kr) for English language editing.

## Author Contributions

**Conceptualization:** Hyunwoo Jung, Junhyup Lee.

**Data curation:** Hyunwoo Jung.

**Formal analysis:** Hyunwoo Jung.

**Methodology:** Hyunwoo Jung, Junhyup Lee.

**Supervision:** Junhyup Lee.

**Validation:** Junhyup Lee.

**Visualization:** Hyunwoo Jung.

**Writing – original draft:** Hyunwoo Jung.

**Writing – review & editing:** Junhyup Lee.

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
