## [Decision Letter · Decision Letter 0]

4 May 2021

PONE-D-21-08427

Estimating the effectiveness of national health insurance in covering catastrophic health expenses: Evidence from South Korea

PLOS ONE

Dear Dr. Jun Hyup,

Thank you for submitting your manuscript to PLOS ONE. After careful consideration, we feel that it has merit but does not fully meet PLOS ONE’s publication criteria as it currently stands. Therefore, we invite you to submit a revised version of the manuscript that addresses the points raised during the review process.

Please address the questions and concerns raised by reviewer 1 when revising the manuscript.

We look forward to receiving your revised manuscript.

Kind regards,

M. Mahmud Khan

Academic Editor

PLOS ONE

Journal Requirements:

We note that you have indicated that data from this study are available upon request. PLOS only allows data to be available upon request if there are legal or ethical restrictions on sharing data publicly. For information on unacceptable data access restrictions, please see http://journals.plos.org/plosone/s/data-availability#loc-unacceptable-data-access-restrictions.

2a) If there are ethical or legal restrictions on sharing a de-identified data set, please explain them in detail (e.g., data contain potentially identifying or sensitive patient information) and who has imposed them (e.g., an ethics committee). Please also provide contact information for a data access committee, ethics committee, or other institutional body to which data requests may be sent.

2b) If there are no restrictions, please upload the minimal anonymized data set necessary to replicate your study findings as either Supporting Information files or to a stable, public repository and provide us with the relevant URLs, DOIs, or accession numbers. Please see http://www.bmj.com/content/340/bmj.c181.long for guidelines on how to de-identify and prepare clinical data for publication. For a list of acceptable repositories, please see http://journals.plos.org/plosone/s/data-availability#loc-recommended-repositories.

Reviewers' comments:

Reviewer's Responses to Questions

**Comments to the Author**

1. Is the manuscript technically sound, and do the data support the conclusions?

Reviewer #1: Partly

Reviewer #2: Yes

2. Has the statistical analysis been performed appropriately and rigorously? 

Reviewer #1: Yes

Reviewer #2: Yes

3. Have the authors made all data underlying the findings in their manuscript fully available?

Reviewer #1: Yes

Reviewer #2: Yes

4. Is the manuscript presented in an intelligible fashion and written in standard English?

Reviewer #1: No

Reviewer #2: Yes

5. Review Comments to the Author

Reviewer #1: Thank you for the opportunity to review this manuscript. This study uses a large data set the Korea Health Panel Survey (2011-2016) to estimate the effectiveness of insurance in covering CHE. This is an interesting submission that, if you choose to go further, I would recommend some major work. There are several major issues that need to be addressed.

Introduction

1. The authors should conclude this section with the objective or research questions you will address in this paper. A priori hypotheses are not mentioned at all.

2. It is unclear whether the authors want to compare the traditional CHE calculation method with a new approach or try to report the amount of CHE covered by health insurance using two different approaches.

3. The study fails to address how the findings relate to previous research in this area. The authors should rewrite their Introduction and Discussion to reference the related literature.

4. It is not a commentary/opinion article, but an original article. Some sentences need citations.

Methods

5. In the Study Sample section, the study population should be described, including the overall population composition, age range, and size. Reasons for exclusion should include the numbers and percentages excluded.

6. What was the proportion of missing data? How were missing data handled in the analyses?

7. Why not include the 2017 KHPS data as well? Please justify.

8. It is unclear why household income is used for rankings in the concentration index.

9. Which computer package was used?

10. Which ethics committee(s) approved your project?

Results

11. In the first paragraph, please describe the characteristics of the sample, beginning with details of the response rates. The demography of the sample should be described and usually, this will include a table of basic indices, including age and gender. Please include the simple statistics (characteristics of the sample or univariate statistics) before moving to more complex analyses.

Discussion

12. Overall, the discussion needs some more work. Several more points could be elaborated on, including, what are some of the barriers that contribute to the findings?

13. It would be more helpful to supply how findings contrast to others in the literature, implications of your results, and potential mechanisms that contribute to them. How were your findings novel? How were they similar?

14. A possible issue with attributing disparities to household income is that income may not represent economic status. They may have monies that are not taxed and current income may not reflect accumulated assets. This at least needs a discussion.

15. Make sure each of your objectives and research questions is discussed.

Reviewer #2: Overall, this manuscript is very valuable study for Korea and for other countries to indicate the advantages and important roles to reduce health expenses through national health insurance system. I think the authors support the results through using appropriate statistical analytical methods. However, I found that the authors should change some Korean language to English in discussion section. For example, in the sixth paragraph in the discussion section, the authors used Korean language in the two references. Therefore, I recommend that the study should change them to English. I would like to recommend to accept this study with minor pending revision.

6. PLOS authors have the option to publish the peer review history of their article (what does this mean?). If published, this will include your full peer review and any attached files.

Reviewer #1: No

Reviewer #2: **Yes: **Seokwon Yoon

---

## [Author Response · Author response to Decision Letter 0]

6 Jun 2021

Sincerely thanks for the valuable comments. We could have updated and made the manuscript better thanks to you. We’ve tried hard to follow your recommendations. Also, in response to the criticism that it was not intellectually written in standard English, we corrected the manuscript with the help of the academic editing site Editage.

Please read the revised version of our manuscript, and please do not hesitate to give advice. 

we submitted a separate attachment with specific answers to review comments.

---

## [Editor Report · Decision Letter 1]

22 Jul 2021

Estimating the effectiveness of national health insurance in covering catastrophic health expenditure: Evidence from South Korea

PONE-D-21-08427R1

Dear Dr. Jun Hyup,

We’re pleased to inform you that your manuscript has been judged scientifically suitable for publication and will be formally accepted for publication once it meets all outstanding technical requirements.

Kind regards,

M. Mahmud Khan

Academic Editor

PLOS ONE

Additional Editor Comments (optional):

I agree with the revisions made. Your method of estimating the CHE reducing effect of insurance is quite innovative and will be useful in analyzing other NHI programs.
---

## [Editor Report · Acceptance letter]

26 Jul 2021

PONE-D-21-08427R1 

Estimating the effectiveness of national health insurance in covering catastrophic health expenditure: Evidence from South Korea 

Dear Dr. Lee:

I'm pleased to inform you that your manuscript has been deemed suitable for publication in PLOS ONE. Congratulations! Your manuscript is now with our production department. 

Kind regards, 

on behalf of

Dr. M. Mahmud Khan 

Academic Editor

PLOS ONE